# MeetingMate: an Ambient Interface for Improved Meeting Effectiveness and Corporate Knowledge Sharing

Justin Matejka[α], Tovi Grossman[μα], and George Fitzmaurice[α]

[α]Autodesk Research and [μ]University of Toronto

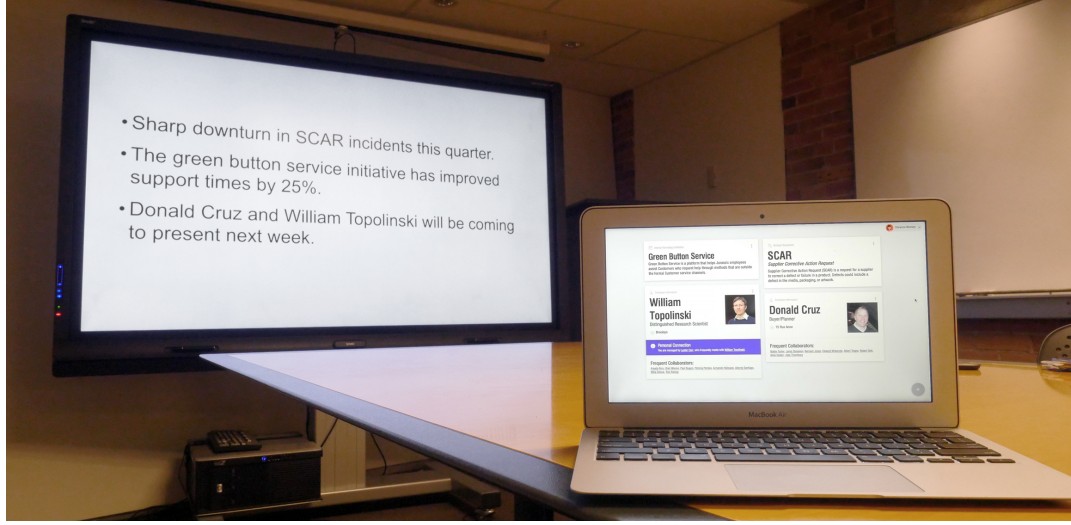

Figure 1. The *MeetingMate* system. The content being presented is captured and interpreted, then relevant corporate knowledge information is displayed on the devices of meeting attendees.

## ABSTRACT

We present *MeetingMate,* a system for improving meeting effectiveness and knowledge transfer within an organization. The system utilizes already existing content produced within the organization (slide decks, meeting information, HR databases, etc.) from which it generates and presents contextually relevant information in real-time to meeting participants through an ambient interface. Besides providing details about projects and content within the company, an employee relationship graph is created which supports increasing a user's "metaknowlege" about *who knows what* and *who knows whom* within the organization.

## 1 INTRODUCTION

The institutional knowledge of a corporation is an important resource [33], and for a corporation to be successful it is necessary for this knowledge to be shared and transferred from those who have it, to those who need it [10]. However, large workforces, distributed locations, and demanding schedules act as barriers to successful knowledge transfer. Companies often employ specific activities designed to improve knowledge sharing such as email newsletters, wiki pages, and all-hands presentations, however, these require employees to do additional work beyond their normal job functions, for some unknown, and unsure, future benefit.

[1] justin.matejka@autodesk.com

[2] tovi@dgp.toronto.edu, tovi.grossman@autodesk.com

[3] george.fitzmaurice@autodesk.com

Besides improving knowledge and awareness of what is going on within a company, it is valuable to improve knowledge of "who knows what" and "who knows whom" within an organization. Such knowledge is referred to as *metaknowledge* [26], and increases in metaknowledge have been linked to improved work performance [27], improved ability to create new innovations combining existing ideas [14], and reduced duplication of work [11].

In knowledge-based work environments it is common for workers to spend between 20% and 80% of their time in meetings [19, 22, 30, 36], and while meetings are considered important [7, 18], they are also often deemed by the attendees to be inefficient and ineffective [20, 29].

This paper describes *MeetingMate*, a system for improving meeting effectiveness and knowledge transfer in an organization through an ambient interface. The MeetingMate system utilizes already existing content produced within the organization as source material (slide decks, meeting information, HR databases, etc.) from which it generates and presents contextually relevant information in real-time to meeting participants. This work contributes a novel technique for extracting presented meeting content directly from an HDMI stream, and is unique in its goal of presenting not only corporate "knowledge" about topics within the company, but also improving employees' "metaknowledge" about *who knows what* and *who knows whom* within the organization.

## 2 RELATED WORK

### 2.1 Meeting Assistance

The development of technology to support and enhance meetings has long been a popular topic of research [38]. Rienks et al. [28] summarize much of the work in "pro-active" meeting assistants, and divide systems into categories based on when they provide assistance: before the meeting, during the meeting, or after the meeting.

Meeting assistants which record the audio and/or visual content of meetings for future viewing are often referred to as "Smart Meeting Systems", and include projects such as the CALO Meeting Assistant System [37] which distributes the task of meeting capture, annotation, and audio transcription, and work by Geyer et al. [8] exploring the idea of allowing meeting participants to create markers into the meeting timeline while the meeting is occurring to improve later navigation. For a more thorough listing of work on "after the meeting" assistance, see Yu and Nakumura [42].

Of the systems designed for in-meeting support, many of them make use of an audio channel. SmartMic [41] makes use of smartphones to capture the audio of a meeting, and the AMIDA system [24] uses microphones in an instrumented meeting room to listen for key words in the conversation of a meeting and pull up or suggest contextually relevant documents. The Connector [5], uses the audio and video channels of a smart meeting room to determine if someone is available to receive a message, and provides mechanisms to deliver the message using the meeting room facilities. Our system is similar in some ways to AMIDA in that both bring up relevant content based on meeting context, however while AMIDA uses the audio of a meeting, our system derives context from the material being sent to the meeting room's projector. We are unaware of any prior work which extracts the visual content being presented in a meeting as context for a real-time meeting assistant.

## 2.2 Corporate Knowledge

Some consider knowledge to be a company's "greatest asset" [35]. Lee et al. developed the KMPI metric [3] to measure how well an organization performs in the area of Knowledge Management measured in five dimensions: knowledge creation, knowledge accumulation, knowledge sharing, knowledge utilization, and knowledge internalization. Our system aims to primarily improve knowledge sharing and knowledge utilization.

For making better use of existing corporate knowledge resources, Zanker and Gordea [43] created a recommendation engine to help when manually searching through internal documents. Aastrand et al. [1] propose using open data to bootstrap the process of creating a hierarchical tagging structure for internal content, while Chen [4] looks at the process of text-mining through corporate documents to extract useful information. When these projects consider searching through and mining corporate data, they are considering "purposefully" created artifacts such as documents and web pages. Our work differs in that while we do mine purposefully created materials such as slide decks and project pages for data, we also make substantial use of "ancillary" corporate data such as meeting room records, mailing lists, and HR databases to generate a more complete picture of the corporate network.

## 2.3 Ambient Information Systems

Ambient interfaces [2, 16, 23, 39] can be characterized as systems which support the monitoring of noncritical information with the intent of not distracting or burdening the user. Ambient displays have been studied for many uses, including software learning [15], social awareness [6], and office work [13].

Pousman and Stasko [25] outline four dimensions in the design of an ambient display system: information capacity, notification level, representational fidelity, and aesthetic emphasis. In our system we are aiming for high information capacity and representational fidelity, while keeping distractions to a minimum with a low notification level.

## 3 CORPORATE KNOWLEDGE CHALLENGES

This work was developed at Autodesk using Autodesk internal data. Autodesk is a multinational software company of ~11,000 employees. The workforce is widely distributed, with many distinct offices, and 17 of those offices house more than 150 employees.

The company faces many of the challenges with corporate knowledge management [21], and results from the yearly employee survey suggest employees generally wish they had more awareness of what is going on in other parts of the company. Autodesk has started a number of initiatives designed to improve awareness and knowledge sharing throughout the company such as wiki pages, project groups, mailing lists, and all-hands presentations. However, since these all require employees to do some additional work beyond their normal job duties without the guarantee of a particular future benefit, these initiatives have not had the desired effect on corporate knowledge sharing.

The goal of our work is to take advantage of the vast amount of material already being produced, and information naturally available within an organization to improve efficiency and awareness. A primary design objective for the system is that there is no additional cost for someone to use the system; that is, it should be just as easy to use the system as it is to not use the system.

## 4 MEETINGMATE

There are many different times and activities through the day where employee corporate knowledge could be improved. We've chosen to focus on times when employees are attending meetings. Since employees are involved in a large number of meetings [19, 22, 30, 36], a system designed to augment the experience of attending meetings would have a broad reach within the organizations, and since those meetings are often considered ineffective [20, 29], a meeting augmentation system could have the dual benefit of increasing overall corporate knowledge, while simultaneously improving the effectiveness of the meeting.

To this end, we've created *MeetingMate*, a system consisting of three main components: a *Data Collector*, a *Presentation Capture System*, and an *Ambient Assistant* (Figure 2).

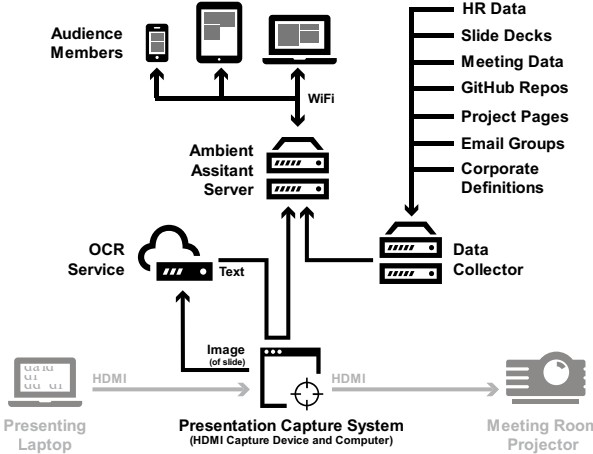

Figure 2. Architecture of the MeetingMate system. (components in light grey are part of the existing meeting room infrastructure).

At a high level, the MeetingMate system uses the visual content presented at meeting as the "search query" for a corporate knowledge database, and presents contextually relevant information to the meeting attendees through an ambiently updating interface. We next describe the three main components of the MeetingMate system in more detail.

## 4.1 Data Sources/Data Collection

This section describes a number of existing data sources within the organization, what information is available within these sources, and how the sources are processed to extract their content. For this

work we only considered data which was "publicly" available within the company, that is, data which everyone within the company has access to. By only using "publicly" available data, we minimize the risk that someone using *MeetingMate* will see privileged or confidential information to which they should not have access.

### 4.1.1 Slide Decks (S1)

Within the company, there are two main locations where documents are stored: a Microsoft Sharepoint [31] server, and an Autodesk A360 project management system. Between the two locations, there are 13,688 PowerPoint (PPT) slide decks dating back to 1997, with 5,343 presentations created between 2014 and 2016. The decks cover a wide range of topics and have been submitted by authors in all divisions of the company.

Processing the slide decks involves two main steps: collecting them from the servers, and analyzing the slides to extract relevant data. For the documents hosted on the Sharepoint server, the Sharepoint API [32] was used to search and download all files of either *.ppt or *.pptx file type. The JFile server does not have a useful API for this purpose, so a web-scraper was written in Python which iterates over each project and crawls through each sub-folder in the documents tree downloading *.ppt and *.pptx files. For both the Sharepoint and JFile based slide decks, high-level metadata such as the creation date, author, and file location are captured during the collection process.

Once the PPTs are downloaded, a data extraction process begins. A C# program using the `Office.Interop.PowerPoint` libraries saves images of each slide in multiple resolutions as .png files, and the text on each slide is extracted and saved to a database.

To download the full collection of 13,688 slide decks and extract the content from the 310,554 slides takes approximately 48 hours on a desktop computer. On a daily basis the Sharepoint and JFile systems are respectively searched and crawled, and newly added PPTs are downloaded and processed. This daily process takes approximately one hour.

### 4.1.2 Meeting Information (S2)

Since the system is restricted internally-public data, we cannot access the calendars of individual employees for meeting records. However, the majority of meetings take place in meeting rooms, which have shared calendars. As the company uses a Microsoft Outlook mail and calendar system, a C# program using the `Office.Interop.Outlook` libraries was written to first collect a list of all meeting rooms, and then step through each of the past meetings which have occurred in the room. For each meeting we record the meeting's: *name* (which often indicates the topic of the meeting), *location*, *length*, and a list of *attendees*.

In total there were 719 meeting rooms which held a total of 355,233 meetings between 2014 and 2016. Collection of the entire data set took approximately 36 hours. The process of accessing each of the individual calendars is relatively time-consuming, taking ~5 hours for incremental daily updates.

### 4.1.3 Code Repositories (S3)

The source code developed by the company is primarily managed through and internal GitHub Enterprise Server. Using a Python script with the GitHub API [9], data for the 6,251 internal git repositories are collected including: *repository name*, *description*, *contributors*, *languages used*, and *bytes of code*. 1,131 employees are listed as contributors to at least one git repository

Data collection for the full set of repositories requires ~3.5 hours. Incremental updates are not easily captured using the API, so the full set of repository data is collected each day.

### 4.1.4 A360 Project Pages (S4)

The A360 project management system is organized into individual "projects" which represent specific working or interest groups within the company. The system houses 3,405 groups, with a median member count of 8. The same crawler used for collecting the PPTs from A360 is used to collect the project information, collecting information such as: *project name*, *project description*, and a list of *group members*.

### 4.1.5 Individual Human Resources Data (S5)

Each of the 11,615 employees (contingent and full-time) at the company has an entry in the internal employee search system. This data is also available in spreadsheet form with 42 columns of information for each employee. Among the most relevant ones are *name*, *email address*, *work location*, *job title*, and *manager's name*. From the *employee name*, and *manager's name* fields we are able to construct the formal organizational structure of the company. Headshot photos (which are available for 58% of employees), follow a consistent naming pattern and location, and are easily downloaded and associated with the appropriate record.

Updating the individual HR data entails copying the daily spreadsheet from the HR system and running the script to look for and download any new, or updated, headshots. This process takes approximately 30 minutes.

### 4.1.6 Email Group Memberships (S6)

To simplify sending emails and meeting requests to collections of people, the company makes use of email groups. There are a total of 15,054 email groups stored on the Microsoft Outlook mail server, with between 1 and 4,316 members in each, with a median member count of 6.

The email group data (*group name*, and *membership list*) is again collected with a C# program using the `Office.Interop.Outlook` library, and the collection completes in approximately 30 minutes.

### 4.1.7 Corporate Definitions (S7)

Stored on the company intranet is an employee-maintained database of acronyms and terms frequently used within the organization. 322 acronyms and 776 terms are defined in this database which is downloaded on a daily basis.

## 4.2 Live Presentation Capture

In order to supplement the presentation material with relevant information, the MeetingMate system needs to be aware of what is being presented. One possible way to do this would be to write an extension for PowerPoint which uses the `Interop.PowerPoint` APIs to extract the data being presented and transfer that information to the MeetingMate server. However, this approach has a number of shortcomings. First, it would only work for presentation material from Microsoft PowerPoint. Second, and more significantly, it would require presenters to do the additional work of installing a plug-in on the machine from which they are presenting. Since a primary design concern of MeetingMate is to *not* require additional set-up work for people to make use of the system, this approach is undesirable.

Our approach is to instead use an HDMI capture and pass through device (designed for live streaming video games) to capture a copy of exactly what is being displayed on the presentation screen. In this way, the presenter performs the exact same steps to present content as they usually (plug a video cable into their laptop), but rather than the cable going directly to the projector, it goes to the HDMI capture device, which passes the signal on to the projector (Figure 2).

The content saved by the HDMI capture device is an image of what is currently being sent to the presentation screen. This image needs to be processed to find any text being displayed.

The windows computer connected to the capture device uploads screenshots to the Project Oxford OCR [17] service for text extraction. The process of uploading the screenshot to the OCR server and receiving the extracted text takes an average of 2.0 seconds. Images are only sent to the OCR service when the projected slide has changed, and this is accomplished by comparing the most recent image with the previously uploaded one, and only uploading the new image if at least 15% of the pixels have changed. The extracted text is sent to the Ambient Assistant server (Figure 2).

Using OCR for the text extraction not only allows for text within images to be recognized, but also enables content from any source (PDF, video, etc.) to be analyzed. This makes MeetingMate completely agnostic to the format of the presented material.

## 4.3 Ambient Assistant

The final piece of the MeetingMate system is the *Ambient Assistant server*, which receives the extracted text from the content being presented, finds relevant corporate knowledge content, and serves the results as a responsive webpage. The server is written in Python with the Flask framework, and a Tornado server running on a Windows Server 2012 instance.

The goal of the Ambient Assistant webpage is to be as unobtrusive and non-disruptive as possible, while still providing useful information which will enhance the audience's understanding of the presentation.

The Ambient Assistant displays relevant knowledge content and is shown on the served page as a series of 'cards' (Figure 1). The individual cards are designed to present the most relevant information at a glance, without requiring input, or too much attention, from the user. As new cards become available they slowly fade in at the bottom of the screen (over a period of 5 seconds) while the page automatically scrolls to make the most recent cards visible. This webpage could be viewed on a number of devices – we have explored several including projecting the ambient assistant onto a secondary screen beside the main projection screen – but believe the most useful configuration is for individuals to view the Ambient Assistant on a personal device such as a phone, tablet, or laptop.

The following sections describe the types of cards which are available, when they are displayed, and what information they contain.

### 4.3.1 Acronyms and Definitions

Corporate communications are often riddled with acronyms and jargon, making the text unnecessarily difficult to understand [34, 40]. As an example, in the 13,688 slide decks collected from the company servers, there are over 132,588 instances of acronyms on the 300,000 slides. However, only 1.7% of those acronyms are defined within the slide deck where they are used.

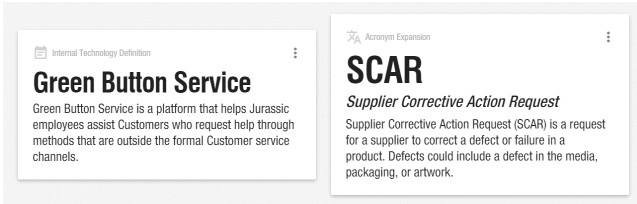

Figure 3. Sample *internal technology definition* (left) and *acronym expansion* (right) cards.

The first two card types are *internal technology definitions* and *acronym expansions*. The presentation text is searched for any of the collected corporate definitions or acronyms (S7), and if they are

found, a card is shown with the acronym expanded (if applicable) and the term defined (Figure 3).

### 4.3.2 Employee Information

The *employee information* card is displayed whenever an employee's name or email address is found on a slide (Figure 4). The lists of employee names and email addresses are derived from the human resources data (S5). Early testing revealed a common occurrence where the name an employee goes by is different from their 'official' name in the HR database (*ex,* Jon Smith *vs.* Jonathan Smith). To overcome this, a list of common name alternatives was used to generate a list of possible names for each person (*ex,* Jonathan Smith could be either "Jon Smith" *or* "Jonathan Smith")

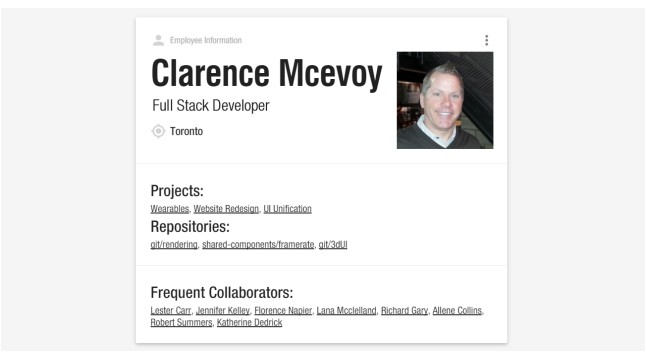

Figure 4. Sample *employee information* card.

Besides the employee's name, the card also displays the employee's headshot, job title, and work location. Additionally, a lists of the projects (S4) and code repositories (S3) the employee is actively contributing to are included. Finally, the most closely related employees using the computed employee network graph (discussed below) are presented as a list of "Frequent Collaborators". Combined, these lists give an overview of both *what* and *who* this employee knows.

### 4.3.3 Projects/Code Repositories

The next two card types are *project* cards and *repository* cards (Figure 5), which are derived from the JFile (S4) and GitHub (S3) data sources. For each, the name and description of the project/repository is displayed, along with names of members or contributors, and in the case of *repository* cards, the list of programming languages used are also shown.

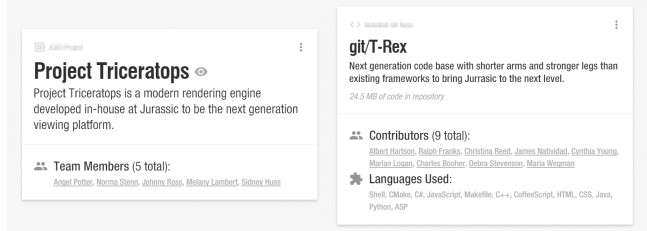

Figure 5. Sample *project* (left) and *repository* (right) cards.

The *project* and *repository* cards are shown whenever the exact project or repository name is found on a slide. Additionally, these cards are displayed if the text of the slide contains many of the same keywords as the description of a project or repository. This is determined by transforming the descriptions of the projects and repositories into *tf-idf* vectors [12], computing the cosine similarity between the descriptions and the recognized text, and displaying the projects/repositories with a cosine similarity > 0.85.

### 4.3.4 Contextually Similar Slides

The final card type displays *contextually relevant slides* (Figure 6) from the collection of over 300,000 slides gathered from the internal slide deck repositories (S1). Besides an image of the contextually relevant slide itself, the card also presents the author of the slide deck, when it was created, and a list of employees who are mentioned somewhere in the deck. Clicking on the "more" icon at the top right of the card presents options to: generate an email containing a link to the contextually relevant presentation, open the presentation directly, or dismiss this card and have it no longer be suggested. The other card types each have similar menus.

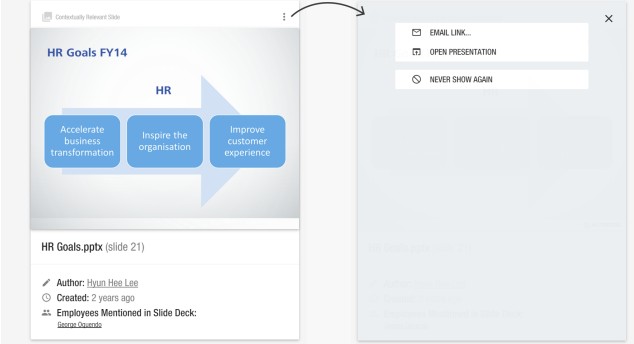

Figure 6. Sample *contextually similar slide* card (left), and the associated context menu (right).

Analogous to the process used to find contextually similar projects and repositories, the text for each of the slides in the slide deck repository are converted to *tf-idf* vectors and compared to captured text. To increase exposure to a wider range of content, if there are any slides with a cosine similarity > 0.85, one slide is chosen at random and displayed.

### 4.3.5 Personal Connections, Employee Relationships

The information on the cards above is tailored to the content being presented, but does not change based on who is viewing the Ambient Assistant. By taking into account who is using the Ambient Assistant ("Clarence Mcevoy" in Figure 7), an additional "Personal Connection" section can be inserted into the cards which highlights a connection the logged in user has to a person or project (Figure 7).

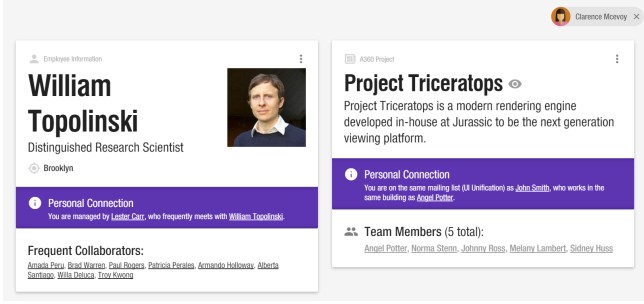

Figure 7. Sample cards with additional *personal connection* information.

This *personal connection* field is populated by looking at an employee network graph computed using the: Meeting Information (S2), Code Repository (S3), A360 Project Pages (S4), HR Data (S5), and Emails Lists (S6) data sets. For each data set, an adjacency matrix is computed based on the strength of the inter-personal connection for each pair of employees.

For example, using the Meeting Information data set, we look at each meeting both employees were a part of, and calculate the weight of the edge between two employees ($e_1$ and $e_2$) with the following formula:

$$\text{weight}_{e1,e2} = \sum_{\text{meetings}(e_1,e_2)}^{m} \left( \frac{length_m}{\#attendees_m} \right)$$

Where *length$_m$* is the length of the meeting, *and #attendees$_m$* is the number of people in the meeting. This means longer meetings, and meetings with fewer attendees are weighted more heavily. Once all edge weights are computed, they are globally ranked from strongest to weakest, and re-mapped between 0 and 1. Similar calculations are carried out for the repository, project page, and email list data sets. For employee data, edges are created between employees and managers.

The edges from the individual data set adjacency matricies are then combined into one overall adjacency matrix by summing the weights from the individual edges to create a composite "weight" metric incorporating all the different measures of adjacency. Then, to find the strongest "personal connection" between any pair of employees, we look for the heaviest-weight, shortest path. That is, we take the set of all shortest paths between the two employee nodes, and then select the path in which the edge weights are the highest.

The heaviest-weight, shortest path is then converted to a natural language sentence by looking at the strongest components of each edge. For example, *"You are managed by Lester Carr, who frequently meets with William Topolinski"*, or *"You are in some project groups (including UI Unification) with Alberta Santiago, who reports to Norma Stenn"*. The overall adjacency matrix represents a single connected component, so a path can be computed between any two employees. However, only paths with at most one intermediate node are displayed to emphasize "strong" connections.

### 5 FEEDBACK AND DISCUSSION

To preliminarily test the design and usefulness of MeetingMate, the system was deployed and tested over a series of weekly group meetings. In several cases the system provided truly unknown information to the audience (in one case while sharing a research paper, an *employee* card for one of the authors showed up – prior to that, attendees at the meeting did not know the author had been hired). This deployment also indicated some ways the system produced spurious or unnecessary results – for example, the internal dictionary contains a definition for "User" as "someone who uses our software", and this definition appeared whenever the (very common) word 'user' appears on a slide. While it is easy for users to mark content to "never be shown again", it would be useful to reduce the number of low-utility results at the system level. In the future more advanced language modelling could look at the surrounding context of the slides, or recent slides, to better predict what results might be most relevant. It would also be interesting to explore recognizing content other than text, such as headshots of particular images, and using those as contextual cues.

The next step is to deploy the system in a meeting room continuously for several months to collect more feedback about how the system performs and is received. This wider deployment will bring up some interesting issues. For meetings concerning highly sensitive topics, we plan to have a very clear, physical switch for presenters to "disable" MeetingMate if they are uncomfortable with the content of their presentation being captured. It will be interesting to see how frequently that functionality is utilized.

While we are limiting ourselves to "publically" available internal data, there are still cases where information which is technically visible to all employees, probably shouldn't be. For example a meeting name could be "Discuss firing John Doe". The creator of

such a meeting is probably unaware that the meeting name is publicly viewable. Practically, the system will need a way to remove these sorts of sensitive entries, but hopefully the deployment of this system will make people more aware of what is visible to other employees. It would also be interesting to make use of non-public information in the system to allow for more personalized recommendations.

Overall, we believe *MeetingMate* serves as a valuable solution for improving meeting effectiveness and knowledge sharing within an organization, and will serve as an example for further work in this area.

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
