# OpenReview forum: "MeetingMate: an Ambient Interface for Improved Meeting Effectiveness and Corporate Knowledge Sharing"
_graphicsinterface.org/Graphics_Interface/2021/Conference/Second_Cycle — GI 2021_

### Official Review · Reviewer_xL5S · 2021-04-19
**A well-written "system case study" short paper.**

**Rating:** 7
**Confidence:** 4

**Review:**

This is a well-written shorter 6-page paper (likely even shorter in the GI format) describing a system that creates a personal ambient display for people to use during company meetings.  It shows company information that is related to the content of the current slides presented in a physical meeting. The method is quite clever, it performs OCR on slide images captured through an HDMI ‘capture and pass-through device” installed in the meeting room. Based on the details in the system description and a preliminary deployment in a series of weekly meetings, this looks like a fully implemented end-to-end system including everything from the capture and OCR, to importing/scraping corporate information from several diverse systems, to the ambient display running as a web application.

This research and paper are polished. The writing is very clear and concise, the related work very effectively explains how the system extends or differentiates from previous work, the system is explained in reasonable detail, and the “Feedback and Discussion” section reports on an initial somewhat informal user validation of the system as a small deployment, and discusses current limitations and future work topics are valuable. I enjoyed reading this paper and I definitely learned something about the topic and I appreciated the system design.

I would classify this as a system paper, maybe more accurately a “system case-study paper” if such a category exists. There is value in reporting this kind applied work, and I think the community would also learn something from how it is built and the problem it’s designed to solve. A longitudinal user study might have even have made this a CHI-level contribution, but I suspect the pandemic restrictions prevented this system, which is designed for physical meetings, to be deployed and evaluated. I hope the authors are planning such a study when the world becomes normal again. I could also imagine the OCR slide scraping methods to be adapted for video conference “Zoom” meetings as well. This could be something to add to the discussion.

Overall, I’m positive about this paper even though it's more like a case study or system paper. Hopefully there’s room for this kind of work at GI.

Other

I’m glad a video was submitted, but unfortunately it adds almost nothing to what’s already n the paper.  I was hoping for a demo of the system actually running in a meeting, and to show more diverse examples of slides and cards than what’s in the paper. It’s important to show a working demo so the reader know the system is really implemented and to better understand how the system looks from a user perspective. A longer demo from a simulated meeting where the slides change, and the related “cards” pop up would have been perfect (e.g. like a time lapse with strategic pauses).  Ideally this would show two or more devices to show ambient display personalization too.  The current video seems to show the exact same figures as the paper for all the UI examples, and even the system view appears to be a faked fade-in of the web UI using the same Figure 1 image. I was quite disappointed with the video, but I think it could be easily improved for a published version.

There’s a typo in 2nd paragraph of related work, I’m assuming “indies” should be “indices”?

---

### Official Review · Reviewer_1Ltn · 2021-05-03
**Interesting idea, needs validation**

**Rating:** 5
**Confidence:** 4

**Review:**


This paper presents a new system (MeetingMate) that aims to make meetings more effective and improve organizational knowledge transfer by projecting additional information on an ambient interface. The system works by issuing implicit search queries based on content being discussed in a meeting and extracts and displays information relevant to the current context (without requiring further interaction from the meeting participants).

MeetingMate appears to be a useful system which has some novel elements for improving meeting participation. I like the idea of having a real-time meeting assistant that can extract helpful and relevant information from ancillary data documents and show them in context. The authors have carefully thought about a diversity of data sources for creating their system, ranging from slide decks to code repositories to corporate definitions. The live presentation capture system using HDMI seemed to be a clever idea and makes the approach perhaps more generalizable. The Ambient Assistant also seems to be flexible in that people can explore this extra information in real-time across many types of devices.

Although the paper is proposing an interesting system idea and the paper is well-written overall, I do not feel that this paper is ready for publication. I have a couple of main concerns:

1.	Lack of validity: A key issue with the paper is that it jumps to explaining the system design (without any cleardesign goals/ formative study) and leaves us hanging at the end without any clear validation. We do not know why this is the right way of tackling this challenge and whether or not this system was actually effective. The authors do mention that they did informally deploy and test the system over a series of group meetings, but I was surprised to see why this was not reported as a user study more systematically? It doesn’t seem like finding participants would be an issue given that they were able to gather informal feedback so they could have at least done a case-study style evaluation.  I do agree that doing a wider deployment is a great idea in the long run, but I did expect to see some insights into users’ perception of usability and usefulness based on their initial interaction. With this, I am not confident about the novel contributions proposed in the paper.

2.	Contextualizing the contribution: Another issue with the paper is that although some related works are mentioned and some differences are noted, I felt that the RW was somewhat shallow. There is plenty of literature in HCI/ CSCW that has investigated interactive ways of improving effectiveness in meetings, but there is no clear articulation in the paper around what is the novel design and research contribution of MeetingMate. I do suspect that there is some new design contribution here with the combination of different data sources and the design of the ambient assistant, but I am not convinced by the writing that this is a new idea and whether or not this is the right way of solving the problem of creating more awareness in meetings.

3.	The discussion is currently a bit shallow. It would be helpful if the paper more thoroughly discussed the implications around privacy and security (some are briefly stated). Again, a user study would be helpful to delineate such issues. There also needs to be some consideration of the generalizability of the design beyond the specific organization that was used.

I hope the authors will continue to improve and revise their paper and submit it in the next cycle.

---

### Official Review · Reviewer_rdPo · 2021-05-03
**Interesting work-in-progress that needs a user study**

**Rating:** 5
**Confidence:** 4

**Review:**

The paper presents a detailed description of a system design to increase effectiveness of corporate meetings and knowledge sharing. The system as well as the technical discussion are sound. My only concern is on the lack details of the preliminary study. Details are missing for the presented preliminary study: how many group meetings, how many people, how long, who used the system, who and how observed the meetings, did the authors interviewed the participants...  The paper is very technical and an opportunity have been missed to learn how participants in corporate meeting would benefit from the proposed design. Without this information the scientific value of the paper is dramatically limited. Authors report on the plan for a longitudinal study, which I am sure would provide interesting findings.

I am wondering whether the system ranks the information according to the user who is interacting with the ambient assistant. In other terms, is the ambient assistant aware of the user and therefore provide personalized information?

Overall the paper is easy to ready, well structured and the description of the system is clear. However, without a discussion of a well-design user study the contribution is limited.

---

### Meta-Review · Area_Chair_Busk · 2021-05-04

**Recommendation:** Accept
**Confidence:** 2

**Metareview:**

Two reviewers are marginally below acceptance because there is no large formal user study. But this is currently very difficult or impossible to do in most (all?) places in the world due to legal and health-related restrictions, especially a study focused on corporate in-person meetings.  It is my opinion that building and deploying the system has already validated the concept to a certain extent, and the paper does report on a smaller informal study which provides a reasonable initial user-centred validation.

I am not entirely sure if GI is willing to accept a short system case study paper like this (this is why I mark my confidence as "no sure"). But bracketing that, I am recommending accept.

If accepted, it's very important the authors make camera-ready revisions to address comments from all reviewers.  Here are the main items I pulled out:
- add details on preliminary user study, ideally with user-centred observations
- improve discussion of related work to further differentiate the novelty of the system
- add to the discussion about privacy and security, etc.
- create a new video that actually demonstrates the system

---

### Decision · Program_Chairs · 2021-05-08

Accept